# Daytime and Overnight Joint Charging Scheduling for Battery Electric Buses Considering Time-Varying Charging Power

**Feifeng Zheng** [1], **Zhixin Wang** [1,*], **Zhaojie Wang** [1] **and Ming Liu** [2]

1   Glorious Sun School of Business and Management, Donghua University, Shanghai 200051, China; ffzheng@dhu.edu.cn (F.Z.); 13761885208@163.com (Z.W.)
2   School of Economics and Management, Tongji University, Shanghai 200092, China; mingliu@tongji.edu.cn
*   Correspondence: wzx921009@163.com

**Abstract:** This work investigates the joint daytime and overnight charging scheduling problem associated with battery electric buses (BEBs) at a single charging station. The objective is to minimize the total charging costs of all BEBs. Two important factors, i.e., peak–valley price and time-varying charging power, are considered to depict real-world charging situations. We establish a mixed-integer programming model for the considered problem, and then conduct a case study together with sensitivity analysis. Numerical results show that compared with the existing first come, first serve rule-based charging solution, the charging schedule obtained by solving the established model via the CPLEX solver can save 7–8% of BEB charging costs. Hence, our model could be applied to improve the BEB charging schedule in practice.

**Keywords:** charging scheduling; battery electric bus; peak and valley electricity price; time-varying charging power; state of charge

## 1. Introduction

In recent decades, the applications of battery electric buses (BEBs) have developed rapidly in many cities all over the world. BEBs play an active role in reducing carbon emissions. Benefiting from their eco-friendly features, many governments have implemented relevant subsidy policies to promote the electrification of public transportation systems [1–3]. Some metropolises in China have taken the lead in popularizing and applying BEBs. By the end of 2017, for example, Shenzhen had achieved a 100% share of BEBs in the urban transportation network. Zhengzhou had also fully replaced fuel buses with new energy buses, and BEBs accounted for 40% by the end of 2020 [4]. Many international cities, such as London, Chicago, and Washington, also used BEBs in their urban transit systems many years ago [5].

Although BEBs have significant merits in energy consumption and environmental protection, the mileage anxiety caused by battery technology constraints is a serious obstacle in the promotion of BEBs [6]. Compared to fuel buses, the charging of BEBs usually takes a relatively long time, which weakens the flexibility of their operational management [7,8]. BEBs mainly use overnight centralized charging to replenish electrical energy. Due to the long charging times and limited charging pile resources, overnight charging may not meet the energy demands of all BEBs in daytime operations [9]. Thus, some BEBs need to return to charging stations for short-time recharging between trips during the daytime. In overnight and daytime charging, peak–valley electricity prices and grid power limitations have critical impacts on BEB charging schedules and costs. A reasonable joint daytime and overnight charging schedule can help mitigate charging activities during peak hours [10,11]. This would satisfy the electricity demand of the bus transit system at a lower cost, as well as improve the utilization of charging piles [12].

Motivated by the above observation, this work attempts to optimize the charging schedules of BEBs with a given bus timetable. Our main contributions are as follows.

First, we introduce the problem associated with the daytime and overnight joint charging scheduling of BEBs. Secondly, a mixed-integer programming model that considers both grid power constraints and peak–valley electricity prices was established. The model can be solved via the CPLEX solver, which outputs an optimal BEB charging schedule. Thirdly, we conducted a case study to reveal the merits of the obtained charging schedule in charging costs, compared with the existing charging solution in practice. The optimal charging schedule can help the decision maker improve the charging pile allocation, start time, and BEB charging duration.

The remainder of this work is organized as follows. In Section 2, we summarize previous studies related to our work. Section 3 provides a detailed description of the considered problem. We present the mathematical model in Section 4. Section 5 presents a case study and sensitivity analysis. Finally, Section 6 concludes this work and provides future research directions.

## 2. Literature Review

There are several research areas related to the daytime and overnight joint charging problems of BEBs, including battery electric vehicle charging scheduling, optimization of BEB charging station locations, BEB charging scheduling, etc. In the following, we present a brief review of the above three problems.

### 2.1. Battery Electric Vehicle Charging Scheduling

In recent years, scholars have conducted various research studies in the field of battery electric vehicles (BEVs), such as BEV energy consumption analysis [13,14], sitting problems of BEV charging stations [15–17], and the optimization of BEB charging scheduling [18]. Ahn et al. (2011) [19] developed an optimal decentralized charging control algorithm for BEVs connected to smart grids, using the flexible characteristics of charging interruption to reduce the power load of the grid. Sundstrom et al. (2012) [20] explored the flexible charging strategy of BEVs, considering the grid power constraints, and proposed a new method for the BEV charging schedule by combining the impact of voltage and power on the grid. The strategy mitigates congestion in the distribution network while meeting the changing needs of BEVs. Flath et al. (2014) [21] pointed out that the charging demand for large-scale BEVs might greatly increase the power load of the grid, and an appropriate coordinated charging strategy could effectively reduce the impact of high power demand on the grid. Although BEV scheduling and BEB scheduling have many similarities, there are fundamental differences in operation management: (1) the former have irregular driving routes and flexible charging locations, while the latter have fixed itineraries and charging stations; and (2) the former make charging decisions based on different origins and destinations of the BEVs, while BEBs follow their fixed travel routes when making charging schedule decisions. Therefore, compared with BEVs, the charging scheduling of BEBs has stronger constraints and a higher level of homogeneity in charging demands.

### 2.2. Optimization of BEB Charging Station Locations

The optimization of BEB charging station locations has a large impact on the daily charging schedules of BEBs. There are many studies on the charging station locations and infrastructure planning of bus transit networks [22–24]. Xylia et al. (2017) [25] developed an optimization model for Stockholm's bus network in order to ensure the availability and efficiency of the charging infrastructure. Wei et al. (2018) [26] optimized the deployment of the BEB system from the perspective of spacetime characteristics, minimizing the total costs of BEB procurement and charging station allocation. Lin et al. (2019) [27] studied the planning problems associated with large-scale BEB charging stations, considering multi-level networks and grid power constraints; the objective was to minimize the total construction costs. An (2020) [28] employed the infrastructure optimization of BEBs under uncertain demands, and considered the impacts of battery capacity, weather factors, and road traffic conditions on the optimization solutions. Wang et al. (2022) [29] proposed a

combinatorial optimization model for the charging pile configuration and fleet scheduling of BEBs, aiming to determine the optimal battery capacity and fleet sizes of BEBs. Ferro et al. (2023) [30] studied the joint optimization problem associated with the location and capacity of BEB charging stations with the objective to minimize the total operating costs. The above studies on the planning and deployment of BEB charging infrastructure focus on decision-making on a strategic level. The results can provide useful references for the construction of urban BEB infrastructure.

*2.3. BEB Charging Scheduling*

In the daily operation of BEBs, there are three basic types of electrical energy replenishment: overnight centralized charging, daytime opportunity charging, and battery exchange [31,32]. Overnight centralized charging means that BEB charging activities occur at charging stations during non-operating hours at night. With battery swapping, a BEB can replace its low-charge battery with a fully charged one at a battery swap station in minutes. Opportunity charging usually happens in the daytime between trips of a BEB; it often takes a shorter time but comes with a higher charging cost compared to overnight charging.

A few authors considered the scheduling problem associated with overnight centralized charging. Houbbadi et al. (2019) [33] proposed an overnight charging schedule that takes into account the impact of battery degradation costs on the charging scheme. Zheng et al. (2022a) [34] developed a set of optimal charging schedules for the overnight centralized charging of BEBs while considering peak–valley electricity prices and battery degradation costs. The proposed schedule effectively mitigates charging activities in peak hours and saves charging costs. In their study, the authors neither considered the time-varying power of charging piles nor daytime charging. Zheng et al. (2022b) [35] considered the uncertain charging times of BEBs in the overnight centralized charging scheduling problem; their objective was to minimize the expected total charging costs. They proved the NP-hardness of the problem and developed a scenario reduction-based enhanced sample average approximation (eSAA) approach, as well as an improved genetic algorithm, to solve large-scale instances. The authors focused on overnight charging and did not consider opportunity charging and the impact of the peak–valley price. Jahic et al. (2019) [36] optimized the BEB centralized charging schedule to balance the high electricity costs in the daytime peak period and load peak constraints.

Some studies focus on the opportunity charging of BEBs. Abdelwahed et al. (2020) [32] investigated the opportunity charging scheduling of BEBs, and developed two optimization frameworks based on discrete time and discrete events. The authors assumed that all BEBs were initially fully charged and that the power of all charging piles remained constant. The work did not consider the impact of grid power constraints on charging schedules. He et al. (2020) [37] considered the impacts of electricity demand costs on the opportunity charging schedule, and proposed a network modeling framework for optimizing the BEB charging schedule. Casella et al. (2021) [38] proposed an optimal BEB charging schedule based on the bus demand response plan under the grid load constraint. They did not consider the impact of the peak–valley electricity price and the variable charging power of the charging schedule. Zhang et al. (2021) [39] studied a BEB opportunity charging scheduling problem that considered battery loss costs and nonlinear charging. The problem was described as a nonlinear programming model, which was solved by the branch and bound algorithm. The authors assumed that once a BEB was charged, it must be charged to its initial state of charge. Liu et al. (2021) [12] considered the BEB charging scheduling problem with real-time power considerations of charging piles, and proposed a column generation algorithm to solve the problem. They assumed that all BEBs had a fixed charging duration; however, BEBs may require different charging times in the real-world based on their remaining electricity power levels.

There are only a few results on the topic of battery exchange. Li et al. (2014) [40] proposed a BEB schedule in the battery swap mode with the objective of minimizing the

total operating costs. Chen and Song (2018) [41] compared two charging methods: opportunity charging and battery swapping. Comparison results showed that battery-swapping technology can mitigate peak power consumption, while it requires more investment in fixed assets, such as buying more batteries. An et al. (2019) [42] balanced the high electricity cost, which is caused by electricity consumption during peak hours, by embedding battery-swapping technology into the local electric bus system.

For the problem associated with daytime and overnight joint charging scheduling, results are scarce in the literature. Abdelwahed et al. (2020) [32] and Liu et al. (2021) [12] are two relevant works that adopted one-day decision-making cycles. However, both of them assumed that all BEBs are fully charged at night; that is, there are no overnight charging scheduling issues involved in their models. In this work, we investigate the daytime and overnight joint charging scheduling of BEBs, i.e., the time horizon of one charging schedule covers a whole day. We consider the impacts of two critical factors, i.e., the peak–valley price and time-varying power of the charging pile on the charging schedule. We aim to provide a charging schedule with minimal total charging costs that determine the real-time power of each charging pile and reduce the charging activities in peak periods under the grid power constraint.

## 3. Problem Description

In this section, we introduce the various elements involved in a single charging station within a bus transit network from the perspective of daily operations, and then provide a detailed explanation of each element.

### 3.1. Bus Transit Network with a Single Charging Station

We use a single charging station within a bus transit network to define a daytime and overnight joint charging scheduling problem for a BEB, minimizing the total charging costs under the constraints of the bus operation timetable. The bus transit network consists of one charging station and several bus lines. The charging station is responsible for providing BEB charging service for all bus lines. Figure 1 illustrates the single charging station bus transit network where there are several BEBs on each bus line. Let $J$ be a set of all the BEBs indexed by $j$, i.e., $j \in J$, in the network. There are $N$ charging piles in the charging station, and each charging pile is equipped with only one charging connection port. Therefore, at most, $N$ BEBs can be charged simultaneously at any time in the charging station.

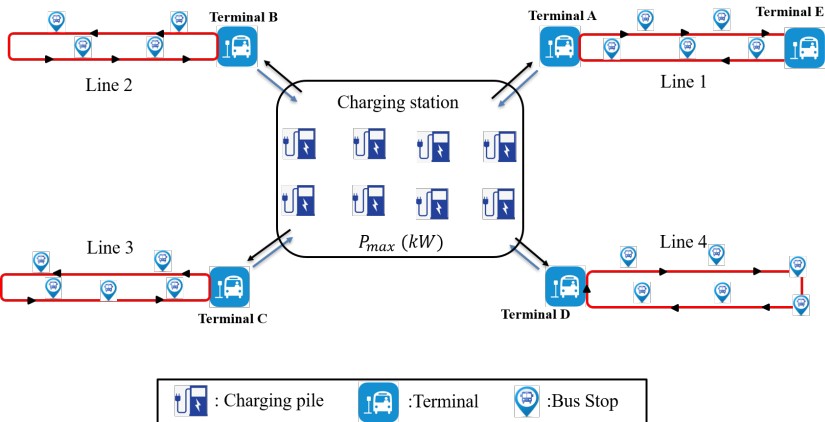

**Figure 1.** Illustration of the bus transit network with one charging station.

The charging station is equipped with an intelligent charging control system, which controls the real-time output power of each charging pile and displays the real-time state of charge (SOC) of any BEB being charged. Let $p_{jt}$ represent the actual charging power of BEB $j$ at time slot $t$, and $P_{max}$ denote the allowable maximum power output of the grid at the charging station. In the setting with time-varying charging power, the value of $p_{jt}$ may vary over time but it is not allowed to be larger than the maximum power $p_a$ allowed

by the battery or the maximum output power $p_b$ of the charging pile. Moreover, the total output power of all $N$ charging piles cannot exceed $P_{max}$ at any time, although it may be true that $N \cdot p_b > P_{max}$; if the total output power of the charging station exceeds the grid power limit $P_{max}$, it will cause damage to the grid and affect the operation safety of the charging station.

Each bus line is composed of one or two terminal stations and multiple bus stops. The BEBs depart from the single charging station in the morning and travel to the dedicated terminals to perform the scheduled transportation tasks. When a BEB has a charging demand, it travels from the corresponding terminal to the charging station for charging. After charging, it returns to the terminal to continue its transportation tasks. Daytime charging activities can only occur during idle time between two adjacent trips. That is, a BEB can only be charged when it stays at the terminal near the charging station, and the charging activity cannot affect its transportation tasks. As shown in Figure 1, for example, a BEB on line 1 can only be charged when it stops at terminal A, although there are two terminal stations. Notice that in the daytime, the BEB can only charge after it arrives at the terminal that is near the charging station. The time duration of the stay (or charging) follows its operation timetable, and it is usually different between different bus lines, for example, the bus lines whose trips are one way or two ways. Since the terminal is close to the charging station, the travel time between the charging station and the terminal is out of consideration in this work [12,43]. When a BEB completes its daily transportation tasks, it goes to the charging station for overnight concentrated charging. Due to the limited number of charging piles in the charging station, overnight centralized charging may not guarantee that all the BEBs are fully charged. Therefore, some BEBs require opportunity charging in the daytime.

*3.2. Peak–Valley Electricity Price*

Due to the gradual increase in the pressure of the electricity demands in most cities, governments usually adopt the peak–valley electricity price method to balance the electricity pressure between different periods. Increasing the cost of electricity during the peak period of electricity consumption can effectively mitigate the impact of centralized electricity on the power grid. As high-powered electrical equipment, charging piles have non-ignorable impacts on the power grid when they are used intensively, especially during peak periods of electricity consumption. For BEB operators, in order to reduce electricity costs and the impact on the power grid, BEB charging activities are usually scheduled during valley periods. Due to the long charging times of BEBs and limited charging pile resources, however, some BEBs may have to be charged during peak periods in order to meet their power demands.

Table 1 presents the partition of the peak–valley electricity periods in Shanghai, China. One can see that there are two peak periods and three parity periods in a day, while there is only one valley period that shows up at night.

**Table 1.** Peak-to-valley period timetable.

|  | Peak Period | Parity Period | Valley Period |
| --- | --- | --- | --- |
| Time interval | 08:00~11:00<br>18:00~22:00 | 06:00~08:00<br>11:00~18:00 | 22:00~6:00<br>- |

Table 2 presents the peak–valley electricity price list in Shanghai. We can see that the electricity price in the peak period is more than three times the valley price, and one to two times that of the parity period. Therefore, the peak–valley electricity price is a critical factor that affects the total charging costs of BEBs. Under the premise of ensuring the daily operation of BEBs, using the concept of avoiding peaks and filling in valleys can effectively reduce the total charging cost of the bus system.

**Table 2.** Peak–valley  electricity price.

| Electricity Classification | | Electricity Price (CNY/kW·h) | | | |
|---|---|---|---|---|---|
| | | Less than 1 kV | 10 kV | 35 kV | More than 110 kV |
| Commercial, industrial electricity | Peak period | 1.074 | 1.049 | 1.024 | 0.999 |
| | Flat valley period | 0.671 | 0.646 | 0.621 | 0.596 |
| | Valley period | 0.316 | 0.310 | 0.304 | 0.298 |
| Agriculture, residential electricity | Peak period | - | 0.730 | - | - |
| | Flat valley period | - | 0.448 | - | - |
| | Valley period | - | 0.242 | - | - |

### 3.3. BEB Operation Timetable

In the bus transit network, BEBs perform transportation tasks according to a pre-set operation timetable. To guarantee the service quality of the bus transit network, the operation timetable is not allowed to be violated. Therefore, the BEBs can only perform their charging activities under the constraints of the operation timetable. The timetable is a time-related description that can be expressed in different ways, and the time-extended network is one of the ways to express the operation timetable in detail. In this work, we discretize the time of day (i.e., an operation cycle) into time slots measured in $\Delta t$; let $T$ denote the set of time slots, indexed by $t$, $t \in T$. We use the 0-1 parameter list $L_{jt}$ to depict the BEB state at each time slot. $L_{jt} = 1$ indicates that BEB $j$ is in the terminal at time slot $t$ (i.e., the BEB is available for charging at the time), and $L_{jt} = 0$ means that the BEB is on the trip. Figure 2 illustrates the operation timetable of BEB $j$. The binary variable $x_{jt} = 1$ denotes that BEB $j$ is charging at the terminal at time slot $t$, and 0 otherwise.

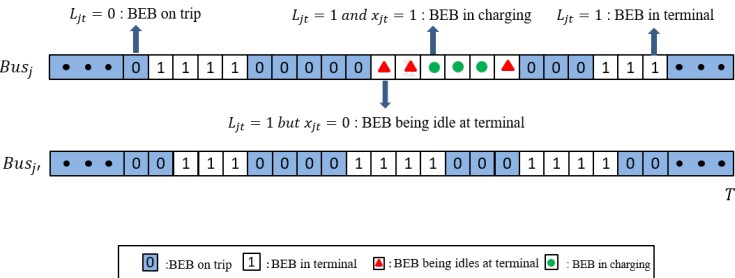

**Figure 2.** Illustration of a BEB operation schedule.

In this study, we focus on minimizing the total charging costs of the BEBs under the constraints of the BEB operation timetable and the grid power limit. A feasible charging schedule needs to decide the start time, charging duration, and pile assignment for each BEB charging activity.

## 4. Model Formulation

In this section, we first introduce the notations of sets, parameters, and decision variables involved in the considered problem. We then formulate a MILP model and explain the constraints of the model.

### 4.1. Notations

The sets, parameters, and decision variables used in the model are defined as follows:
**Sets:**

$J$: Set of BEBs, indexed by $j$, i.e., $j \in J = \{1, 2, ..., |J|\}$;
$T$: Set of times, indexed by $t$, i.e., $t \in T = \{1, 2, ..., |T|\}$;

**Parameters:**

$L_{jt}$: Location of BEBs, 1 if BEB $j \in J$ is at the terminal at time slot $t \in T$;

$f_t$: Electricity cost at time slot $t \in T$ in CNY/kWh;

$C_j^{int}$: Initial SOC of BEB $j$ in kWh;

$C_{min}$ Minimum state of charge required by BEB $j \in J$ during daytime operations in kWh;

$C_{max}$ Maximum battery capacity in kWh;

$p_a$: Maximum power allowed by the battery in kW;

$P_b$: Maximum transmit power of each charging pile in kW;

$e_t$: Electricity consumption per time slot during the operation of a BEB in kWh;

$\Delta t$: Duration of each slot;

$P_{max}$ Total output power, upper limit of the charging station in kW;

$N$: Number of charging piles in the charging station;

**Decision Variables:**

$x_{jt}$: Binary variable, equal to 1 if BEB $j \in J$ is charging at time slot $t \in T$, 0 otherwise;

$I_{jt}$: Binary variable, auxiliary variable for controlling the continuity of charge between time slot $t$ and $t + 1$, $I_{jt} \in \{0,1\}$;

$V_{jt}$: Binary variable, auxiliary variable for controlling the continuity of charge between time slot $t$ and $t - 1$, $V_{jt} \in \{0,1\}$;

$C_{jt}$: Continuous variable, SOC of BEB $j \in J$ at time slot $t \in T$ in kWh;

$\Delta C_{jt}$: Continuous variable, amount of SOC variations of BEB $j \in J$ at time slot $t \in T$ in kWh;

$p_{jt}$: Continuous variable, charging power of BEB $j \in J$ at time slot $t$;

$Q$: Continuous variable, total charging cost.

*4.2. Mathematical Model*

Based on the above definitions of the parameters and variables, the following MILP model is established for the considered problem.

$$Q = \min \sum_{j \in J} \sum_{t \in T} p_{jt} \cdot \Delta t \cdot f_t \tag{1}$$

Subject to:

$$p_{jt} = 0 \quad \forall j \in J \quad \forall t \in T \quad L_{jt} = 0 \tag{2}$$

$$0 \leq p_{jt} \leq \min\{p_a, p_b\} \quad \forall j \in J \quad \forall t \in T \quad L_{jt} = 1 \tag{3}$$

$$C_{jt} = C_j^{int} \quad \forall j \in J \quad t = 1 \tag{4}$$

$$C_{jt} = C_{j,t-1} + \Delta C_{jt} \quad \forall j \in J \quad \forall t \in T \tag{5}$$

$$C_{min} \leq C_{jt} \leq C_{max} \quad \forall j \in J \quad \forall t \in T \tag{6}$$

$$\Delta C_{jt} = p_{jt} \cdot x_{jt} \quad \forall j \in J \quad \forall t \in T \quad L_{jt} = 1 \tag{7}$$

$$\Delta C_{jt} \leq \min\{p_a, p_b\} \cdot x_{jt} \quad \forall j \in J \quad \forall t \in T \quad L_{jt} = 1 \tag{8}$$

$$\Delta C_{jt} \leq p_{jt} \quad \forall j \in J \quad \forall t \in T \quad L_{jt} = 1 \tag{9}$$

$$\Delta C_{jt} \geq p_{jt} - \min\{p_a, p_b\} \cdot (1 - x_{jt}) \quad \forall j \in J \quad \forall t \in T \quad L_{jt} = 1 \tag{10}$$

$$\Delta C_{jt} = -e_t \quad \forall j \in J \quad \forall t \in T \quad L_{jt} = 0 \tag{11}$$

$$\sum_{j \in J} x_{jt} \leq N \quad \forall t \in T \tag{12}$$

$$\sum_{j \in J} p_{jt} \leq P_{max} \quad \forall t \in T \tag{13}$$

$$I_{jt} \geq x_{jt} - x_{j,t+1} \quad \forall j \in J \quad \forall t \in T \quad L_{jt} = 1 \tag{14}$$

$$V_{jt} \geq x_{jt} - x_{j,t-1} \quad \forall j \in J \quad \forall t \in T \quad L_{jt} = 1 \tag{15}$$

$$\sum_{t \in T} I_{jt} = \sum_{t \in T} V_{jt} \quad \forall j \in J \quad L_{jt} = 1 \tag{16}$$

$$\sum_{t \in T} I_{jt} \leq 1 \quad \forall j \in J \quad L_{jt} = 1 \tag{17}$$

$$\sum_{t \in T} V_{jt} \leq 1 \quad \forall j \in J \quad L_{jt} = 1 \tag{18}$$

$$x_{jt}, I_{jt}, V_{jt} \in \{0, 1\} \quad \forall j \in J \quad \forall t \in T \tag{19}$$

$$C_{jt}, p_{jt}, \Delta C_{jt}, Q \in \mathbb{R}^+ \quad \forall j \in J \quad \forall t \in T \tag{20}$$

Equation (1) is the objective function used to minimize the total charging cost. In this problem, we focus on the electricity cost, where $f_t$ represents the electricity cost at time slot $t$ and it is related to the electricity price at the slot.

Constraint (2) means that the charging power of a BEB on a trip is 0 because the BEB cannot be charged when executing a transportation task. Constraint (3) represents the possible charging power of the BEB when it is at the terminal or charging station. Constraints (4)–(6) calculate the SOC of the BEB at any time slot. Constraint (7) calculates the SOC variation of each time slot when the BEB is at the terminal or charging station. Constraints (8)–(10) linearize constraint (7). Constraint (11) represents the variation of the SOC of the BEB in each time slot during the trip. Constraint (12) denotes the service capacity constraint of the charging station, which means that the number of BEBs being charged simultaneously cannot exceed the number of charging piles. Constraint (13) ensures the maximum power limit of the grid. Constraints (14)–(18) represent the continuity of the bus during charging. Constraints (19) and (20) denote the value ranges of decision variables.

Considering the peak–valley electricity price and controllable charging power, the BEB charging scheduling problem is formulated as a MILP model. The model can be solved using the commercial solver CPLEX. In the following, we conduct case studies using real bus transit networks in Shanghai, China.

## 5. Case Studies

In order to verify the validity of the model and provide some meaningful management insights into the public transport system, we conducted case studies on two public transport sub-networks in Shanghai. In addition, we also conducted a sensitivity analysis on some parameters that have critical impacts on the BEB charging schedule, revealing how the objective value varies with different resource equipment.

### 5.1. Data Description

In the bus transit network with a single charging station, the charging station only provides charging services for the BEBs on some dedicated lines. The terminals of these

bus lines are distributed near the charging station, and then the travel time between the terminals and the charging station is neglected. In the case study, we use a single-line bus network and a multi-line bus network in the Qingpu district, Shanghai, as two examples. We define a round trip of any BEB as a trip. In Figure 3, for example, a trip on line S1 is defined as traveling from terminal T1 to terminal T2 and then returning to terminal T1 (i.e., $T1 \rightarrow T2 \rightarrow T1$).

The trip of the BEB is fixed, and the position of the BEB at each time point is also predetermined. That is, the BEB strictly follows the operation timetable $L_{jt}$. All BEBs in the bus network are homogeneous, equipped with 240 kWh LiFePO$_4$ batteries. The maximum charging power allowed for the battery $p_a$ is 90 kW. According to the peak–valley electricity price in Table 2, we can derive the electricity price $f_t$ at any time slot $t$. In addition, referring to the research by Liu et al. (2021) [12], we set the power consumption of the BEB to 0.25 kWh per minute during operation. In the case study, we define the length of one time slot as one minute (i.e., $\Delta t = 1$), and one day from 5:30 to 23:30 is divided into 1080 time slots. Therefore, the power consumption $e_t$ of one time slot during the driving of any BEB is 0.25 kWh. We define the start time of each cycle as 5:30 every morning.

### 5.2. Single-Line Bus Network

In the single-line bus network, we consider the case with two charging stations and two bus lines, and each charging station only provides charging services for the BEBs on the dedicated line. As shown in Figure 3, charging stations A and B only provide charging services for BEBs on line S1 and line S2, respectively. Single-line bus networks are generally designed for cross-regional bus lines, which have long mileage. There are usually many BEBs running on the line, and the bus company equips a small charging station for one cross-regional line.

The basic information of the two bus lines, S1 and S2, for the case of a single-line bus network, is shown in Table 3. Line S1 has 18 BEBs, and charging station A is equipped with 4 charging piles. Line 2 includes 17 BEBs, and charging station B is equipped with 4 charging piles. Each charging pile is equipped with a single connector. Each BEB on lines S1 and S2 has, respectively, 3 and 4 trips a day. The maximum transmit power $p_b$ of each charging pile is 80 kW, and the total power limit $p_{max}$ allowed for each charging station is 300 kW.

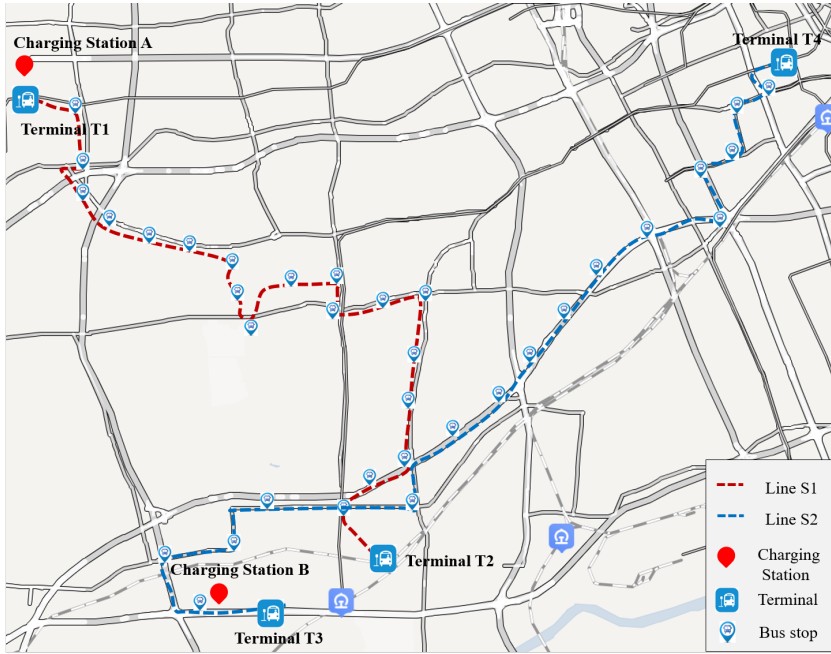

**Figure 3.** The single-line bus network in Qingpu district, Shanghai, in the case study.

**Table 3.** Basic information on the single-line bus network.

| Bus Line | Number of BEBs | Mileage | Number of Trips | Trip Duration |
|----------|----------------|---------|-----------------|---------------|
| Line S1 | 18 | 74.8 km | 3 | 140 min |
| Line S2 | 17 | 65.4 km | 4 | 130 min |

Based on the above parameter settings, we apply the charging schedule developed in this work to the above bus transit network. We solve the MILP model in Section 4 via CPLEX, and obtain an optimal charging scheduling solution. To verify the merits of the obtained charging schedule, we compare the experimental results with the total cost of the existing charging scheme. The existing charging scheduling scheme adopts the idea of the first come, first serve (FCFS) rule and a BEB with the lowest electricity level is of the highest charging priority. Moreover, the existing charging scheme does not take into account the impact of the peak–valley electricity price. The experiments were conducted on a PC with Intel Core i7, 4.0 GHz processors, and 16 GB RAM. The computational time is in CPU seconds.

According to the requirements of the safe operation of the public transport system, the SOC of each BEB cannot be lower than the lower limit $C_{min}$ at any time. Through the investigation of the bus transit network, we set the lower limit of SOC as 35%; that is, the SOC of BEB at any time cannot be less than 35% of the battery capacity. According to the above parameter settings, the experimental results are shown in Table 4.

**Table 4.** Numerical experiment results.

| Bus Line | Objective | Solution Time | FCFS | GAP |
|----------|-----------|---------------|------|-----|
| Line S1 | CNY 1174.12 | 1947.3 s | CNY 1268.91 | 7.5% |
| Line S2 | CNY 1086.78 | 1766.5 s | CNY 1172.37 | 7.3% |

Commercial solvers, such as CPLEX, output optimal solutions for lines S1 and S2 at 1947.3 s and 1766.5 s, respectively. The optimal objective values for the two lines are CNY 1174.12 and CNY 1086.78, respectively. The current existing schedule based on the FCFS rule outputs objective values equal to CNY 1268.91 and CNY 1172.37, respectively. We define GAP as the relative error of the objective value of the FCFS solution compared with that of the optimal one. According to Table 4, the BEB charging schedule derived by solving the MILP model can save 7.3–7.5% of the charging cost.

### 5.3. Multi-Line Bus Network

In the multi-line bus network, the terminals of the bus lines are distributed near the charging station, and then the travel time between the terminals and the charging station is neglected. The multi-line bus network consists of one charging station and four bus lines. The charging station provides charging services for all the BEBs stopping at terminals A, B, and C. All the BEBs are charged overnight at the charging station, and in the daytime, some BEBs with low power levels also require opportunity charging at the charging station. Opportunity charging activities only take place when the BEBs arrive at one of the terminals (A, B, or C). The charging activities must end before starting the following transportation tasks of the corresponding BEBs (Figure 4).

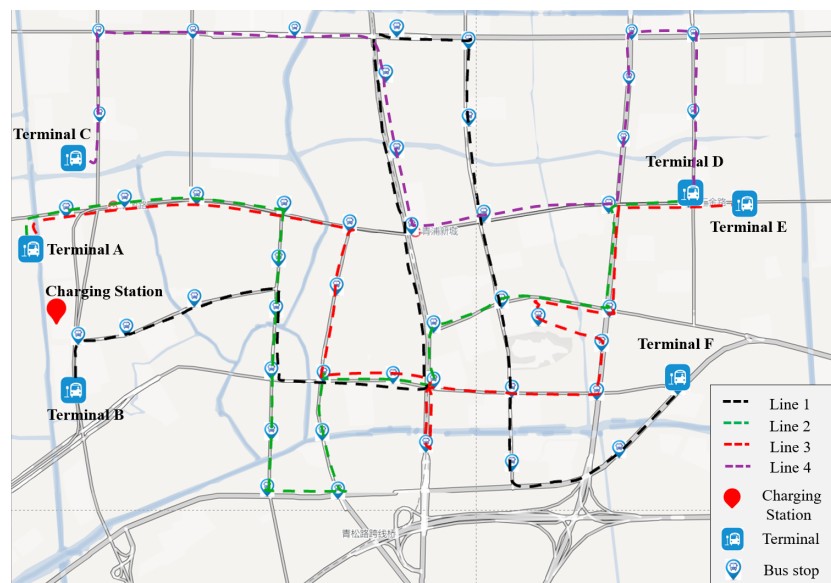

**Figure 4.** The multi-line bus network in Qingpu district, Shanghai, in the case study.

The multi-line bus network has 29 BEBs in total, and all the BEBs are homogeneous, equipped with 240 kWh LiFePO$_4$ batteries. There are 6 charging piles in the charging station, and each charging pile is equipped with a single connector. The maximum transmitting power $p_b$ of each charging pile is 80 kW, and the total power limit $p_{max}$ allowed for the bus transit network is 420 kW. Basic data of the four bus lines are given in Table 5. The operating timetable of each BEB in the multi-line bus network is shown in Table 6.

**Table 5.** Basic data of the multi-line bus network.

| Bus Line | Number of BEBs | Mileage | Number of Trips | Trip Duration |
|----------|----------------|---------|-----------------|---------------|
| Line 1 | 7 | 24.2 km | 7 | 90 min |
| Line 2 | 8 | 28.2 km | 6 | 100 min |
| Line 3 | 7 | 26.6 km | 7 | 90 min |
| Line 4 | 7 | 25.6 km | 7 | 90 min |

**Table 6.** Operating timetables of the BEBs.

| Bus Number | Time Slots on the Trip |
|------------|------------------------|
| Line 1–1 | 5:30–7:00; 7:40–9:10; 9:50–11:20; 12:00–13:30; 14:10–15:40; 16:20–17:50; 18:30–20:00 |
| Line 1–2 | 5:50–7:20; 8:00–9:30; 10:10–11:40; 12:20–13:50; 14:30–16:00; 16:40–18:10; 18:50–20:20 |
| Line 1–3 | 6:10–7:40; 8:20–9:50; 10:30–12:00; 12:40–14:10; 14:50–16:20; 17:00–18:30; 19:10–20:40 |
| Line 1–4 | 6:30–8:00; 8:40–10:10; 10:50–12:20; 13:00–14:30; 15:10–16:40; 17:20–18:50; 19:30–21:00 |
| Line 1–5 | 6:50–8:20; 9:00–10:30; 11:10–12:40; 13:20–14:50; 15:30–17:00; 17:40–19:10; 19:50–21:20 |
| Line 1–6 | 7:10–8:40; 9:20–10:50; 11:30–13:00; 13:40–15:10; 15:50–17:20; 18:00–19:30; 20:10–21:40 |
| Line 1–7 | 7:30–9:00; 9:40–11:10; 11:50–13:20; 14:00–15:30; 16:10–17:40; 18:20–19:50; 20:30–22:00 |
| Line 2–1 | 6:00–7:40; 8:40–10:20; 11:20–13:00; 14:00–15:40; 16:40–18:20; 19:20–21:00 |
| Line 2–2 | 6:20–8:00; 9:00–10:40; 11:40–13:20; 14:20–16:00; 17:00–18:40; 19:40–21:20 |
| Line 2–3 | 6:40–8:20; 9:20–11:00; 12:00–13:40; 14:40–16:20; 17:20–19:00; 20:00–21:40 |
| Line 2–4 | 7:00–8:40; 9:40–11:20; 12:20–14:00; 15:00–16:40; 17:40–19:20; 20:20–22:00 |
| Line 2–5 | 7:20–9:00; 10:00–11:40; 12:40–14:20; 15:20–17:00; 18:00–19:40; 20:40–22:20 |
| Line 2–6 | 7:40–9:20; 10:20–12:00; 13:00–14:40; 15:40–17:20; 18:20–20:00; 21:00–22:40 |
| Line 2–7 | 8:00–9:40; 10:40–12:20; 13:20–15:00; 16:00–17:40; 18:40–20:20; 21:20–23:00 |
| Line 2–8 | 8:20–10:00; 11:00–12:40; 13:40–15:20; 16:20–18:00; 19:00–20:40; 21:40–23:20 |

**Table 6.** *Cont.*

| Bus Number | Time Slots on the Trip |
| --- | --- |
| Line 3–1 | 5:30–7:00; 7:40–9:10; 9:50–11:20; 12:00–13:30; 14:10–15:40; 16:20–17:50; 18:30–20:00 |
| Line 3–2 | 5:50–7:20; 8:00–9:30; 10:10–11:40; 12:20–13:50; 14:30-16:00; 16:40–18:10; 18:50–20:20 |
| Line 3–3 | 6:10–7:40; 8:20–9:50; 10:30–12:00; 12:40–14:10; 14:50-16:20; 17:00–18:30; 19:10–20:40 |
| Line 3–4 | 6:30–8:00; 8:40–10:10; 10:50–12:20; 13:00–14:30; 15:10–16:40; 17:20–18:50; 19:30–21:00 |
| Line 3–5 | 6:50–8:20; 9:00–10:30; 11:10–12:40; 13:20–14:50; 15:30–17:00; 17:40–19:10; 19:50–21:20 |
| Line 3–6 | 7:10–8:40; 9:20–10:50; 11:30–13:00; 13:40–15:10; 15:50–17:20; 18:00–19:30; 20:10–21:40 |
| Line 3–7 | 7:30–9:00; 9:40-11:10; 11:50–13:20; 14:00–15:30; 16:10–17:40; 18:20–19:50; 20:30–22:00 |
| Line 4–1 | 5:40–7:10; 8:00-9:30; 10:20–11:50; 12:40–14:10; 15:00–16:30; 17:20–18:50; 19:40–21:10 |
| Line 4–2 | 6:00–7:30; 8:20–9:50; 10:40–12:10; 13:00–14:30; 15:20–16:50; 17:40–19:10; 20:00–21:30 |
| Line 4–3 | 6:20–7:50; 8:40–10:10; 11:00–12:30; 13:20–14:50; 15:40–17:10; 18:00–19:30; 20:20–21:50 |
| Line 4–4 | 6:40–8:10; 9:00–10:30; 11:20–12:50; 13:40–15:10; 16:00–17:30; 18:20–19:50; 20:40–22:10 |
| Line 4–5 | 7:00–8:30; 9:20–10:50; 11:40–13:10; 14:00–15:30; 16:20–17:50; 18:40–20:10; 21:00–22:30 |
| Line 4–6 | 7:20–8:50; 9:40–11:10; 12:00–13:30; 14:20–15:50; 16:40–18:10; 19:00–20:30; 21:20–22:50 |
| Line 4–7 | 7:40–9:10; 10:00–11:30; 12:20–13:50; 14:40–16:10; 17:00–18:30; 19:20–20:50; 21:40–23:10 |

Again, we use CPLEX, which outputs the optimal solution in 3567.2 s, and the optimal cost is CNY 1824.97. The total charging cost of the existing FCFS charging scheme is CNY 1975.54. Hence, the BEB charging schedule derived by solving the MILP model saves 7.6% of the charging cost.

In order to uncover the reasons that affect the total charging cost, we further explore the total power of the charging station in one cycle. Figure 5 shows the variation of the total output power in the charging station within a single period under the optimal charging schedule generated by solving the MILP model. It can be seen from the figure that under the constraints of the peak–valley electricity price, the total power reaches its maximum during the valley period. Moreover, under the grid power constraint, the total power of the charging station is always less than the upper limit $p_{max}$. Due to the limited resources of charging piles, the total electric power demand of all the BEBs cannot be met during the valley's electricity price period. Thus, some BEBs need to be charged during parity and even peak periods. Notice that the charging price in the peak period is much higher than that of the other two period types. According to the figure, there is almost zero charging power during the peak periods in the optimal charging schedule.

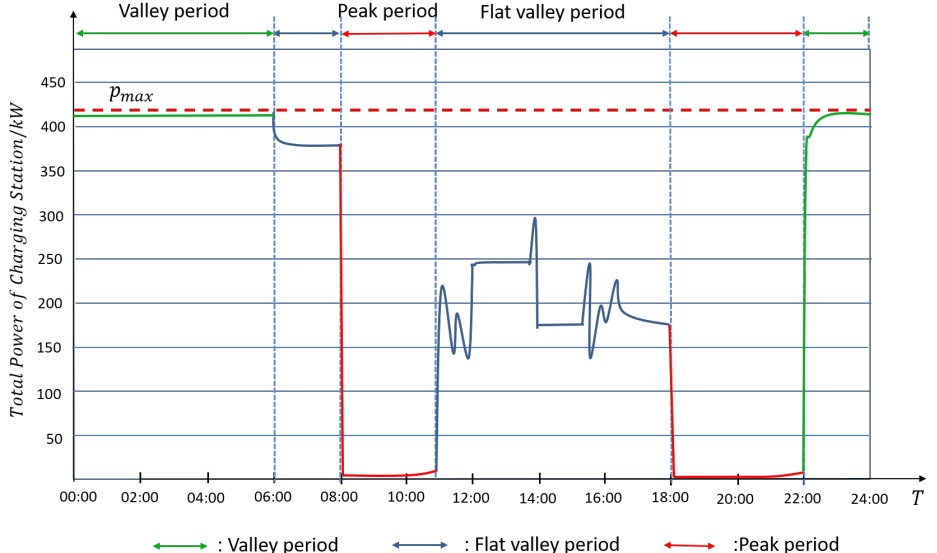

**Figure 5.** Charging power variation of the optimal charging schedule.

Figure 6 shows the charging power variation of the existing charging scheme based on the FCFS rule and without considering the peak–valley electricity price. It can be seen from the figure that when the bus transit network does not consider the peak–valley electricity price, some BEBs will be charged in the peak periods. Similarly, since overnight charging cannot meet the total power demand of the BEBs in a day, opportunity charging activities occur mostly in the idle time slots of the BEBs between their last trips. Note that the opportunity charging activities contain both the parity period and peak period in the afternoon, where the BEBs are more likely to have high electrical power levels. Compared with the optimal charging schedule, the existing charging scheme uses many peak time slots for BEB charging, resulting in a higher charging cost.

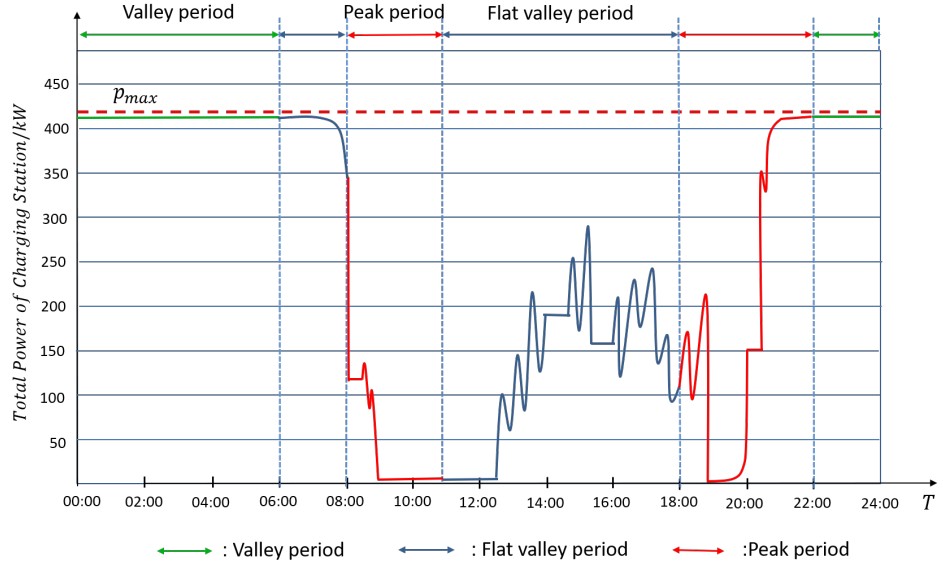

**Figure 6.** Charging power variation of the existing FCFS rule-based charging scheme.

To further explore the SOC changes of BEBs in the optimal charging schedule, we randomly select 4 BEBs, i.e., lines 1–1, 2–3, 3–6, and 4–7 in the bus transit network, and the real-time SOCs of the 4 BEBs are shown in Figure 7. It can be seen in the figure that lines 1–1 and 2–3 are fully charged in the valley periods at night, and their SOCs can meet the power demands of daytime operations. Therefore, both BEBs require no opportunity charging. Due to the limited charging piles, lines 3–6 and lines 4–7 cannot be fully charged in the overnight centralized charging. They need opportunity charging in the daytime. To avoid the high price of peak periods, both BEBs are charged during the parity periods, resulting in the lowest total charging cost.

By observing the output power of each charging pile, it can be seen that each charging pile (on charging) is assigned an obviously higher output power in the valley period than in the peak period. This distribution pattern of charging power allows the charging station to make full use of its charging capacity in the valley period and, thus, minimize the total charging costs of the BEBs. The BEB will only be charged at a peak period when its remaining SOC is insufficient for its next transportation trip. In this case, the BEB is usually charged with a small amount of electricity energy to reduce its charging cost; thus, it is reasonable to allocate a charging pile with a low output power for this purpose.

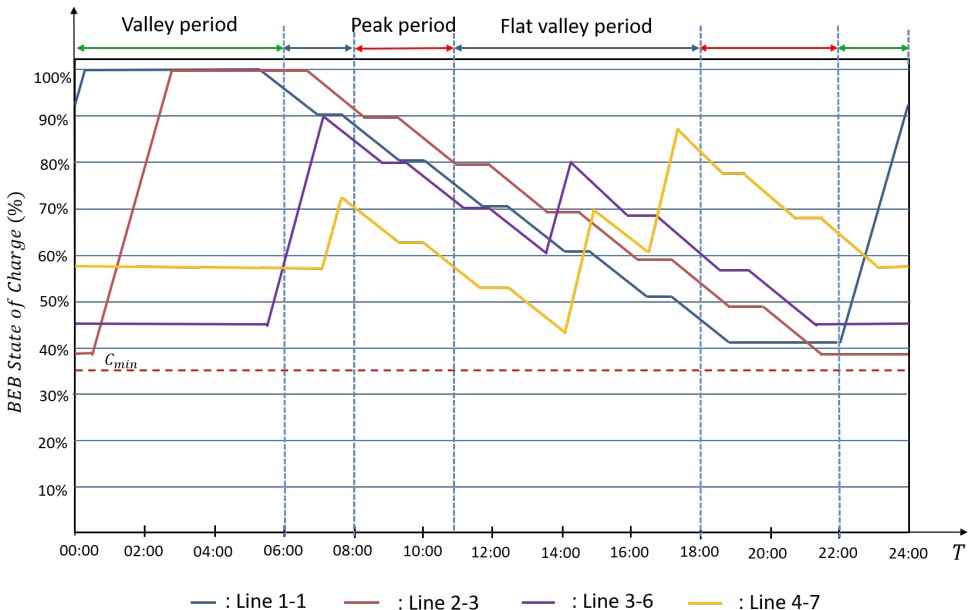

**Figure 7.** Battery SOC variation in the optimal charging scheme.

## 6. Sensitivity Analysis

In order to uncover the impacts of the critical factors on the total charging costs of BEBs, we conduct a sensitivity analysis on the battery capacity, the number of charging piles, and the minimum SOC.

### 6.1. Impact of Battery Capacity

First, we explore the variation of the total charging cost in the optimal charging schedule with different battery capacities of the BEBs. All the other factors, including the number of BEBs and charging piles, remain unchanged. The relationship between the battery capacity and total charging cost is depicted in Figure 8.

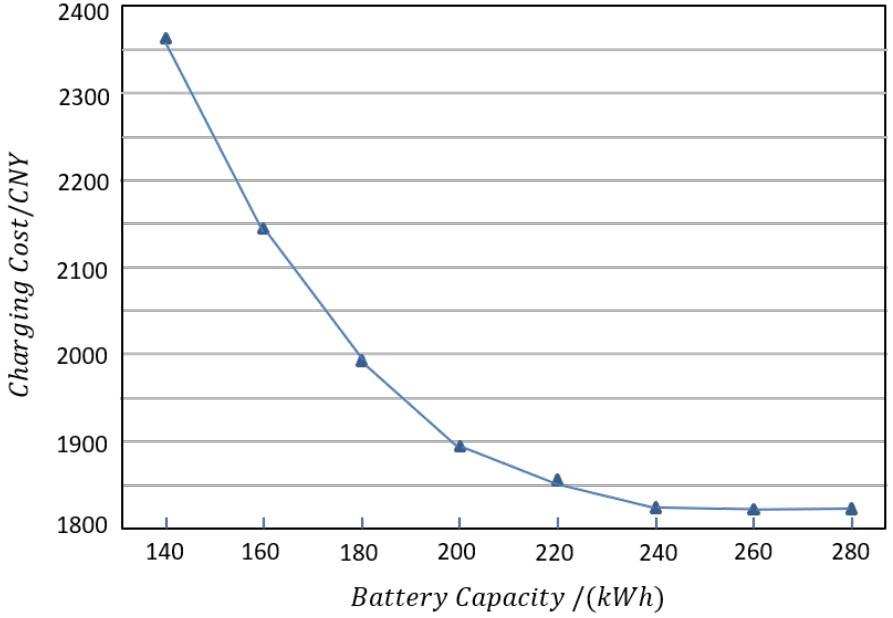

**Figure 8.** Variation in total charging cost with different battery capacities.

It can be seen from Figure 8 that the total charging cost first decreases with the increase in the battery capacity and then remains almost unchanged when the battery capacity is larger than 240 kWh. This phenomenon is because when the battery capacity is low, the SOCs of all the BEBs after overnight charging cannot meet the power demands of their daily operations, even if all the BEBs are fully charged in the valley period at night. Thus, the BEBs have to use opportunity charging during parity or peak periods in the daytime, resulting in high total charging costs. As the battery capacities increase, the BEBs are charged with more electrical power in the centralized charging; then, it is possible to avoid opportunity charging in the peak periods. When the battery capacity is equal to or larger than 240 kWh, the total power demand of all the BEBs can be completely satisfied in the valley and parity periods, resulting in the lowest total charging cost. Notice that the total charging cost does not reduce with the battery capacity for the case where the capacity is at least 260 kWh. In this case, the charging pile resource in the valley period at night has already been fully utilized. Overnight centralized charging cannot meet the daytime power demand of all the BEBs. Some BEBs have to be recharged in the daytime because they cannot be fully charged overnight due to the constraints of limited charging piles and the power grid. Hence, it is helpless to further increase the battery capacity of BEBs since it is not the bottleneck anymore when it is larger than 260 kWh.

### 6.2. Impact of the Number of Charging Piles

Next, we test the impact of the number of charging piles on the total charging cost in an optimal charging solution. Charging piles are one of the key resources in the development of the BEB industry. The construction scale of charging piles directly affects the investment level of BEBs. In order to derive a complete variation trend of the total charging cost with a different number of charging piles, we ignore the constraints of the grid power. We consider the case where the power of each charging pile is 70 kW, and focus on the case where there are 29 BEBs and the uniform battery capacity is 240 kWh. The numerical results are shown in Figure 9.

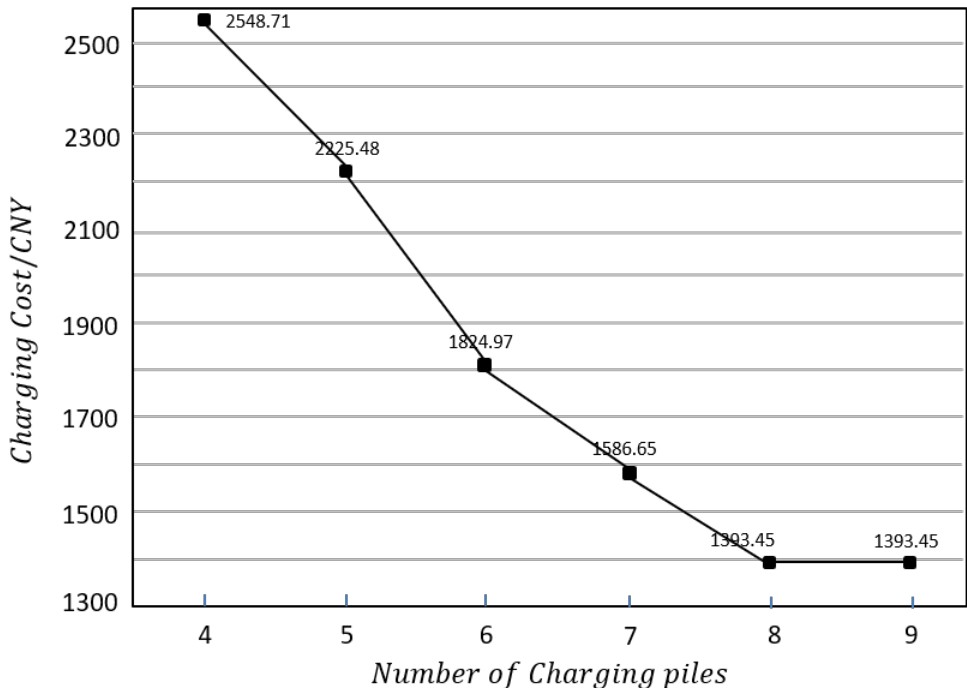

**Figure 9.** Variation in total charging cost with different numbers of charging piles.

Figure 9 shows that when there are only four or five charging piles, the total charging cost is very high. This is because a large proportion of the power demand of the BEBs

cannot be satisfied in the valley periods at night due to the limited number of charging piles. Many BEBs have to use opportunity charging in the parity periods and even peak periods, resulting in a relatively high objective value. As the number of charging piles continues to increase, the total charging cost decreases as more power demand can be satisfied in the valley period with more charging piles. When there are more than eight charging piles, we find that the total charging cost no longer decreases. This is because, with eight charging piles, all the BEBs can be fully charged in the overnight centralized charging; providing more charging piles cannot further improve the SOC state of the BEBs after charging at night.

### 6.3. Impact of the Minimum SOC

The minimum SOC, i.e., $C_{min}$, is a key factor to ensure the safe operation of BEBs. That is, the SOC of each BEB cannot be lower than $C_{min}$ at any time. Based on operation experience, bus companies usually set $C_{min}$ to 35% of the battery capacity. The value of $C_{min}$ not only affects the battery capacity utilization but also the charging schedule and charging cost. Therefore, we test the impact of $C_{min}$ on the total charging cost in the optimal charging schedule. We consider the configuration of the bus network in Section 5.2, and set $C_{min}$ to be 20%, 25%, 30%, 35%, 40%, and 45%, respectively. The experimental results are given in Figure 10.

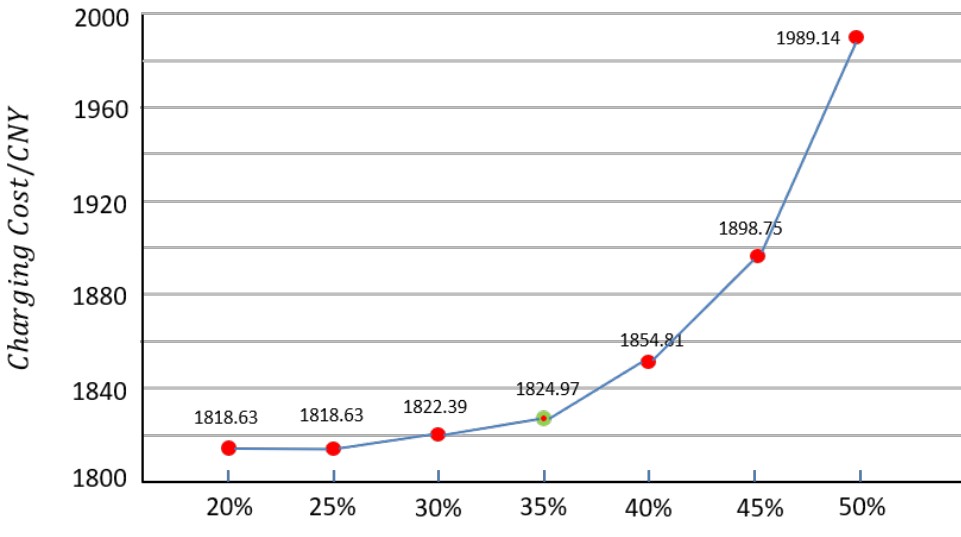

**Figure 10.** Variation in total charging cost with different minimum SOCs.

It can be seen from Figure 10 that when the value of $C_{min}$ increases from 20% to 35%, the total charging cost only slightly increases with $C_{min}$, i.e., an increase of less than 0.35%. However, when $C_{min}$ increases from 35% to 50%, the total charging cost increases by 9.00%. This is because the available battery capacity decreases with $C_{min}$. When the value of $C_{min}$ is relatively small, the merit of overnight charging with the valley's electricity price can be well utilized. For the case with a large $C_{min}$, the electricity energy contribution of overnight charging to daytime operations is weakened, and then BEBs need opportunity charging with parity or peak prices in the daytime. Opportunity charging pushes up the total charging cost. A lower $C_{min}$ implies a larger depth of discharging of the battery. It may lead to more severe battery degradation and decrease the battery life. Therefore, the minimum SOC $C_{min} = 35\%$ in this work is a reasonable setting.

Based on the above analysis, we conclude that under the premise of considering the peak–valley electricity price, grid constraints, and variable charging power, the daytime and overnight joint charging scheduling scheme can effectively reduce the total charging

cost of BEBs. Furthermore, a reasonable configuration of battery capacity, the number of charging piles, and minimum SOC also play vital roles in reducing the total charging cost of the bus transit system.

## 7. Conclusions

This work studies the problem associated with daytime and overnight joint charging scheduling of BEBs. We focus on the impact of the peak–valley electricity price and grid power constraints on the optimal charging schedule, and consider the variable power of charging piles. The daytime and overnight joint charging scheduling of BEBs can effectively utilize the charging pile resources in the current public transportation system, and avoid the issue caused by adopting only centralized charging with insufficient charging resources. The considered problem is formulated as a mixed-integer linear programming model and is solved by the commercial solver, CPLEX. We conducted a case study on a real bus sub-network in Shanghai. The results of the case study show that the charging scheduling scheme proposed in this paper saves 7.8% of the charging cost compared with the existing charging scheme that uses the FCFS rule. Sensitivity analyses further show that the total charging cost decreases with both the battery capacity and the number of charging piles, while it increases with the minimum SOC of the battery.

There are some interesting directions in future research: (1) this study only considered a relatively small bus transit network with a single charging station, and we ignored the travel time between terminals and the charging station. Thus, it may be a meaningful extension to study the charging schedule on a large bus network with multiple charging stations, where the travel times between terminals and charging stations are considered; (2) the battery degradation cost is a hidden cost that is easy to overlook, and the charging scheduling problem considering the battery degradation cost may also be an interesting issue; and (3) the power consumption of a BEB per unit of time is assumed to be constant in this work, while the charging scheduling scheme in the environment with random power consumption will be worthy of investigation.

**Author Contributions:** Conceptualization, F.Z. and M.L.; Methodology, Z.W. (Zhixin Wang); Validation, Z.W. (Zhixin Wang); Formal analysis, M.L.; Investigation, Z.W. (Zhaojie Wang) and M.L.; Writing—original draft, Z.W. (Zhixin Wang); Writing—review & editing, F.Z.; Supervision, M.L. All authors have read and agreed to the published version of the manuscript.

**Funding:** This work was supported by the National Natural Science Foundation of China (grant nos. 72271051, 71832001, 72071144); the Fundamental Research Funds for the Central Universities (grant no. 2232018H-07), and the Fundamental Research Funds for the Central Universities and Graduate Student Innovation Fund of Donghua University (grant no. CUSF-DH-D-2022053).

**Conflicts of Interest:** The authors declare no conflict of interest.

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
