# Peer review of "Daytime and Overnight Joint Charging Scheduling for Battery Electric Buses Considering Time-Varying Charging Power"

_sustainability, doi:10.3390/su151310728_

Round 1
Reviewer 1 Report
BEBs can be charged in the daytime and overnight to meet their daily operational electricity demands. The topic is interesting, please see my comments below:
1) The authors have already published the work in the operational research journal 2022. What is the difference between them?
2) Are you considering a variable battery size to be optimized in the model? If the battery size is assumed to be constant and overnight charging is considered to be complete, how the model works?
3) Travel time from the terminal to the charging station is neglected in this study, why? In reality, it could be significant.
4) Do the authors investigate defining the trip as one way instead of two ways, this may have a high impact on the charging schedule.
Proofreading is recommended.
Reviewer 2 Report
The paper proposes a scheduling procedure for EV bus. the paper is timely and well written. But this paper faces similar topics like Ferro, G., Minciardi, R., Parodi, L., & Robba, M. (2023). Optimal Location and Line Assignment for Electric Bus Charging Stations. IEEE Systems Journal. and CASELLA, V., et al. Optimal charging of electric buses: a periodic discrete event approach. In: 2021 29th Mediterranean Conference on Control and Automation (MED). IEEE, 2021. p. 208-213.
Please double check the overall paper
Reviewer 3 Report
This work studies the problem of daytime and overnight joint charging scheduling of BEBs. They proposed a mixed integer programming model and save charging cost compared with the existing first come first serve rule based charging solution. However, the study would be enriched if more examples of real data were added to verify the validity of the model. The analysis and study of additional cases allows further exploration of the selection of model parameters, the optimisation of the model and the precise range of application of the model. The influence of the different parameters on the final results should also be specified in the article.
The language and grammar of the essay are relatively simple and the language descriptions are clear.
Reviewer 4 Report
The author(s) have clearly explained the problem description and solution to the problem, with the necessary supportive results. This manuscript can be accepted in its current form.
Round 2
Reviewer 1 Report
The authors addressed my comments despite the truth that some comments referred to future work.
Author Response
Dear Editor,
Thank you very much for your efforts on reviewing this paper and your recommendation.
We have now resubmitted the new manuscript to the journal. We hope that the revised version can meet with approval.
Best regards,
Zhixin Wang (on behalf of all the authors)
Reviewer 2 Report
All my comments have been addressed
now it is ok
Author Response

(The authors gave the same response as above.)

Reviewer 3 Report
The revised manuscript has added more literature review and the influence of the different parameters on the final results .However, the study would be enriched if more examples of real data were added to verify the validity of the model. And there is few improvement in the revision.
The language and grammar of the essay are relatively simple and the language descriptions are clear.
Author Response
Dear Editor,
Thank you very much for your great comment. According to your suggestion, we have conducted another case study on the so-called single-line bus network in Section 5, which is different from the multi-line bus network in the previous manuscript. We consider the case with two charging stations and two bus lines, and each bus line is served by a dedicated charging station. Please refer to lines 302-363 on pages 8-10 of the revised manuscript.
We have reorganized the abstract, the introduction section and the literature review section carefully, and double-checked the whole paper. Now we resubmit it to the journal. We hope that the revised manuscript can meet with approval of the journal.
Zhixin Wang (on behalf of all the authors)
Donghua University